# Recent Progress of Biomimetic Tactile Sensing Technology Based on Magnetic Sensors

**DOI:** 10.3390/bios12111054

**Published:** 2022-11-21

**Authors:** Jiandong Man, Guangyuan Chen, Jiamin Chen

**Affiliations:** 1State Key Laboratory of Transducer Technology, Aerospace Information Research Institute, Chinese Academy of Sciences, Beijing 100190, China; 2School of Electronic, Electrical and Communication Engineering, University of Chinese Academy of Sciences, Beijing 100049, China

**Keywords:** biomimetic tactile sensing, magnetic sensor, sensor application, review of technology progress

## Abstract

In the past two decades, biomimetic tactile sensing technology has been a hot spot in academia. It has prospective applications in many fields such as medical treatment, health monitoring, robot tactile feedback, and human–machine interaction. With the rapid development of magnetic sensors, biomimetic tactile sensing technology based on magnetic sensors (which are called magnetic tactile sensors below) has been widely studied in recent years. In order to clarify the development status and application characteristics of magnetic tactile sensors, this paper firstly reviews the magnetic tactile sensors from three aspects: the types of magnetic sensors, the sources of magnetic field, and the structures of sensitive bodies used in magnetic tactile sensors. Secondly, the development of magnetic tactile sensors in four applications of robot precision grasping, texture characterization, flow velocity measurement, and medical treatment is introduced in detail. Finally, this paper analyzes technical difficulties and proposes prospective research directions for magnetic tactile sensors.

## 1. Introduction

Tactile perception is one of the most important ways for organisms to obtain environmental information, just like vision and hearing. How to make robots acquire tactile perception like human beings is one of the hot spots in scientific research. With the idea of bionics, a large number of tactile sensors have been designed based on the working principle of human skin. Biomimetic tactile sensors are important media for robots to perceive external environment, which help robots get information about pressure, vibration, roughness, and temperature. Tactile sensors have played an important role in medical treatment [1,2], artificial skin [3,4], robot tactile feedback [5], and human–machine interaction [6,7]. With the discovery of new materials and the development of microelectronics, tactile sensors based on a variety of transducing mechanisms such as resistance [8,9,10,11], capacitance [12,13,14,15], piezoelectric [16,17,18,19], and optics [20,21,22] have been developed.

In recent years, magnetic sensors are developing towards the direction of high sensitivity, low power consumption, small size, etc. [23]. The performance of some types of magnetic sensors has already met the requirements of tactile sensing technology [24]. Therefore, a new type of tactile sensors named magnetic tactile sensors have recently developed rapidly [25,26,27,28,29]. Compared with tactile sensors based on other mechanisms, magnetic tactile sensors have the advantages of high sensitivity, low hysteresis, low power consumption, easy implementation of three-dimensional detection, and remote detection [30].

However, there are few articles to comprehensively and systematically summarize studies on magnetic tactile sensors so far. Most of the related articles are reviews of tactile sensing technology [31,32,33]. These articles focus on resistive, capacitive, and piezoelectric tactile sensors; furthermore, a few articles which involve magnetic sensors are also very concise and not comprehensive enough [30,34,35,36]. In order to clarify the development status and application characteristics, this paper focuses on tactile sensing technology based on magnetic sensors and summarizes related research in detail. Firstly, according to the types of magnetic sensors, the characteristics of each kind of magnetic sensors used for tactile sensing technology are introduced. Secondly, the sources of magnetic field used in these studies are classified and their characteristics are analyzed. Thirdly, the structures of sensitive bodies used in different magnetic tactile sensors are summarized. In addition, this paper introduces the applications of magnetic tactile sensors, including robot precision grasping, texture characterization, flow velocity measurement, and medical treatment. Finally, the technical difficulties and future prospects of magnetic tactile sensors are analyzed.

## 2. Classification and Development

There are many types of magnetic sensors, including Hall sensors, anisotropic magnetoresistive (AMR) sensors, giant magnetoresistive (GMR) sensors, tunnel magnetoresistive (TMR) sensors, giant magnetoimpedance (GMI) sensors, fluxgate sensors, atomic magnetometers, and superconducting quantum interference devices (SQUID). These types of sensors have different characteristics in tactile perception, so this chapter first classifies magnetic tactile sensors according to the types. The following is a detailed description of the sources of magnetic field and the structures of sensitive bodies used in these studies, which also play a crucial role in the performance of the magnetic tactile sensor.

### 2.1. Types of Magnetic Sensors

By summarizing related literature in recent years, the characteristics and performances of all kinds of magnetic tactile sensors are sorted out in Table 1.

Based on Table 1, the following part of this paper introduces relevant research in detail according to the types of sensors.

#### 2.1.1. Hall Sensor

Hall sensors are based on the Hall effect, which refers to the physical phenomenon that a transverse potential difference is produced when the magnetic field acts on the carriers in metal conductors or semiconductors. The illustration of Hall effect is shown in Figure 1a. The manufacturing process of Hall sensors is mature and compatible with the semiconductor manufacturing process. Hall sensors are the most common magnetic sensors because of their low cost and simple fabrication. They have a wide detection range and are often used to detect strong magnetic fields. Although the performance of Hall sensors is not as good as magnetoresistive sensors in weak magnetic detection, they have been widely used as magnetic switches and magnetometers in various applications [79,80].

Hall sensors are frequently used in tactile sensing. The most common usage method is to insert both permanent magnets and Hall sensors into flexible material. When the flexible material is deformed by force, the position of permanent magnets will change. So, the magnetic strength and direction of the magnetic field around the magnets are changed. By detecting the change with Hall sensors, the amplitude and direction of the tactile force can be obtained successfully. Based on this principle, Eduardo et al. developed a Hall tactile sensor as early as 2006 [37]. They made a hollow hemispherical structure with flexible silicone. A permanent magnet was placed on the top of the hemisphere, with four Hall sensors underneath. Hall sensors detect the position change of the magnet caused by the contact. Finally, the sensor achieved a resolution of 94 mN for normal force.

The commercialization of Hall sensors is very successful, and the enhancement of tactile sensor performance is also closely related to the development of commercial Hall sensors. In particular, the commercialization of three-dimensional magnetic sensors makes the perception of three-dimensional tactile force easier. Ledermann et al. combined a three-dimensional magnetic sensor chip AS54xx (Fraunhofer Institute for Integrated Circuits, Erlangen, Germany) with flexible silicone for tactile sensing [38], so they could measure three-dimensional force with only one sensor. Its minimum resolution is 150 mN with little hysteresis. A classic representative of commercial three-dimensional Hall sensors is MLX90393 (Melexis, Ieper, Belgium), which is widely used by researchers [42,43,44,45,46,81]. For example, Hongbo Wang has made a variety of tactile sensing devices with MLX90393 [42,43]. One of the typical devices is shown in Figure 1b. The MLX90393 chip is under a permanent magnet, and a flexible body separates the chip from the magnet, as shown in Figure 1c. This tactile sensor has excellent performance. It can distinguish the smallest force of 0.71 mN in the X or Y axis and 1.42 mN in the Z axis, which is the best resolution achieved by Hall tactile sensors at present.

As shown in Table 1, there are a lot of studies on tactile sensing with Hall sensors. At present, Hall sensors are able to detect the minimum force on a millinewton scale. However, there are still many problems. For example, the sensitivity of Hall sensors is not high enough, and it is not able to detect the force on a micronewton scale or smaller. To increase the detection capability of Hall sensors, auxiliary structures such as magnetic flux concentrators have also been studied [82]. However, it increases the size of the device, and the fabrication process is much more complex. In addition, the materials commonly used for Hall sensors are greatly influenced by temperature [83], which limits the application of Hall tactile sensors in complex environments.

Benefiting from a large measuring range and low cost, Hall sensors will still be the main magnetic sensor used in tactile sensing in the future. Especially with the development of packaging technology, the ability of Hall sensors to realize three-dimensional perception in small volume has become increasingly prominent, which is of great significance for the development of three-dimensional tactile perception.

#### 2.1.2. AMR Sensor

The AMR effect refers to the phenomenon that the resistance of anisotropic magnetic materials varies with the change of the angle between the magnetization and the current direction, as shown in Figure 2a. The magnetic sensitivity of AMR sensors is improved compared with that of Hall sensors. AMR sensors have low 1/f noise and are more suitable for low-frequency magnetic detection. It is most commonly used for geomagnetic detection.

There are also a small number of reports on tactile sensing with AMR sensors. Ping Yu produced a tactile sensor with a commercial 3-axis AMR sensor (Honeywell HMC1053), as shown in Figure 2b [53]. A permanent magnet was embedded in the flexible hollow PDMS (polydimethylsiloxane) material, and the AMR sensor was below the magnet. Finally, this sensor could detect a minimum force of 10 mN, with a sensitivity of 58 mV/N and a working range of 0–20 N (Z-axis).

AMR sensors are one type of magnetoresistive sensors, which are easier to fabricate than GMR sensors and TMR sensors. Therefore, AMR sensors are easier to combine with other complex processes to achieve more powerful functions. In 2022, Christian Becker et al. combined the AMR manufacturing process with traditional three-dimensional origami technology, and made a three-dimensional device, which could realize three-dimensional perception of magnetic field [55]. The three-dimensional sensitive structure is shown in Figure 2c. By using the origami technology, they can bend the AMR film to a non-horizontal plane. Then, they combined the sensor array with a neodymium iron boron (NdFeB) permanent magnet attached to the end of cilia. The manufacturing process of the sensor is shown in Figure 2d. When the hairs in Figure 2d are bent by force, the position of the permanent magnet will change. By detecting the magnetic strength through the three-dimensional AMR sensor array, they could realize the perception of three-dimensional tactile force. However, due to the limitation of the performance of AMR sensors, this sensor could only detect the magnetic strength on a millitesla scale. Therefore, the large permanent magnet must be used in this study, which is not conducive to achieving a large spatial resolution.

There are not many studies on tactile sensing with AMR sensors. On the one hand, the performance of mature Hall sensors has been able to meet the needs of general tactile sensing. On the other hand, for high-precision tactile sensing, the performance of AMR still lags behind that of other magnetoresistive sensors. In addition, due to the influence of working temperature and external magnetic field, AMR sensors with the Wheatstone bridge structure easily generate bridge bias after being used for a period of time. It is usually necessary to rearrange the internal magnetic domains with auxiliary devices such as set/reset coils. This increases the volume of the sensor and is not conducive to the long-term use of the device.

In the future, AMR sensors should give full play to the advantages of being easily manufactured. AMR sensors can be fully integrated with the existing manufacturing processes of flexible tactile structures to achieve more complex and powerful tactile capabilities.

#### 2.1.3. GMR Sensor

The giant magnetoresistance (GMR) effect is a quantum mechanical phenomenon which was first found in the structure of multilayer magnetic metal film [84,85]. Now, most GMR sensors are spin-valve structures [86]. The most classical structure of spin valve is two layers of ferromagnet sandwiched with a layer of conductive materials (such as Cu, Cr), as shown in Figure 3a. When the magnetic moment directions of adjacent ferromagnetic layers are parallel or anti parallel under the action of external magnetic field, the resistance of multilayers will change. The resistance change of GMR sensors is one order of magnitude larger than that of AMR sensors, so a weaker magnetic strength can be detected. Soon after its discovery, the GMR effect played a great role in magnetic storage, and has great application value in sensors [23].

At the beginning of the 21st century, GMR sensors have been widely used as tactile sensing elements. For example, Goka and Nakamoto have successively designed tactile sensors with GMR elements [57,58,87,88]. Goka used flexible circular bulges embedded with permanent magnets as the sensitive structure. A commercial GMR sensor from the NVE company was used to obtain signals. The working range of this sensor is −40 to 40 N, and the minimum force that can be detected is 60 mN. Nakamoto et al. further optimized this design. They used a magnetic film embedded with two permanent magnets and multiple GMR sensors, and the large-area perception of contact force was successfully realized.

With the help of bionics, the realization of artificial skin is an important research target of tactile sensing technology. The requirement of artificial skin is achieving large-area, high-resolution perception while maintaining flexibility. There are also attempts in this field with magnetic sensors. Jin Ge et al. grew GMR multilayer films on flexible PI substrates as sensitive elements, and successfully produced a thin-film magnetic tactile skin, as shown in Figure 3b [68]. They used permanent magnetic particles as the source of the magnetic field and designed an air gap between the skin and the sensor. When contacting external objects, the magnetic skin is closer to the GMR sensor below, so the magnetic signal is stronger. This sensor realizes non-contact measurement, which has great application value in medical diagnosis. In addition, it is worth mentioning that this study made a pyramid convex structure under the magnetic skin. When the force is large, the top of the pyramid structure will contact the GMR sensor and deform with the increase of the force. This design greatly increases the detection range of applied force. Although this sensor is fully flexible, it has the problem of a reduction of durability. In addition to the fact that the electrode is prone to fracture after multiple bending, GMR multilayer films are also very fragile. Therefore, this sensor is not suitable for applications in complex scenes such as large angle bending and stretching.

In order to balance flexibility and complexity, Miguel Neto designed a flexible-rigid hybrid sensor on a robot finger in 2021 [69]. They produced GMR sensors on silicon wafer and flipped them on a flexible circuit board. The connection between the two was realized by a silver conductive epoxy resin, as shown in Figure 3c. Since the silicon wafer is small enough (0.8 mm × 1.5 mm), the whole device is still flexible enough and is more adaptable to harsh environments.

The commercialization of GMR sensors is later than that of Hall sensors, but GMR sensors develop quickly and have a tendency to replace Hall sensors under many situations. There are still many reports on GMR sensors in tactile sensing in recent years, as shown in Table 1. However, the fabrication process of multilayer films is much more complicated, resulting in a higher cost. Furthermore, compared with Hall sensors, GMR sensors exhibit a large reduction in operating range, so they are prone to saturation when measuring a large tactile force. These are the problems that GMR tactile sensors need to solve in the future.

#### 2.1.4. TMR Sensor

There are many similarities between TMR sensors and GMR sensors, and both of them are multilayer film structures. However, the middle layer in TMR multilayer films is not metal materials, but insulating materials (such as AlOx, MgO). The illustration of the structure of TMR sensors is shown in Figure 4a. The magnetic moment direction of the pinned layer is fixed as a single direction, and the direction of the free layer can follow the direction of the external magnetic field, as shown by the arrows in Figure 4a. When the two directions are the same, the resistance of the multilayers is large, and the resistance is small when they are different. The TMR effect is based on the tunneling effect of electrons in insulating materials, and the change in resistivity is one order of magnitude larger than the GMR effect. Therefore, TMR sensors have extremely high sensitivity, which are very suitable for weak magnetic field detection. At the same time, TMR sensors also have better temperature stability and lower power consumption.

At present, there are only a small amount of studies on tactile sensing with TMR sensors. In 2018, Dongfang Zhang proposed a new tactile sensor for detecting the hardness of objects [70]. As shown in Figure 4b, this sensor is composed of an elastomer, a ferromagnetic probe, a TMR element, and a concave magnet. When objects with different hardness are placed on the elastomer, the probe inside the elastomer moves down at different speeds. Objects with higher hardness make the probe move down faster, resulting in faster changes in the magnetic strength. The TMR sensor they used is a commercial chip (Multidimensional TMR2009). There is another study on tactile sensing with this kind of chip [72]. However, these studies are only preliminary attempts to apply TMR elements in tactile sensing. The performances of the sensors are not excellent, while the volumes of the whole sensors are too large.

Although TMR sensors can theoretically provide better resolution for tactile sensing, there are not many reports. This is mainly due to the influence of 1/f noise [89,90]. When the frequency of the magnetic field is low, the noise of the TMR sensors is very high. For tactile sensing, in addition to the detection of high-frequency vibration, most of the signals are low-frequency. Therefore, TMR is not applicable in this situation. In addition, compared with AMR and GMR sensors, the film thickness of TMR sensors has a significant impact on the performance of the sensor. Furthermore, the current in TMR sensors needs to flow vertically to the film plane, which has much higher requirements for the manufacturing process. The complexity of the process and lower consistency make it difficult for mass production, which is a major obstacle for the wide application of TMR sensors.

Thanks to the high sensitivity of TMR sensors to weak magnetic fields, the future application of TMR sensors in tactile sensing can focus more on the measurement of force on micro-Newton and nano-Newton scales. Additionally, it can be applied in special application areas such as minimally invasive surgery.

#### 2.1.5. GMI Sensor

GMI sensors are designed based on the giant magneto-impedance effect. The giant magnetoimpedance effect refers to the phenomenon that the impedance of specific materials changes significantly with the change of magnetic field. GMI sensors are a kind of magnetic sensor that theoretically integrates many advantages such as high sensitivity, fast response, low hysteresis, wide temperature range, good stability, and low cost. It has the advantages that other magnetic sensors do not have or cannot have at the same time [91].

GMI sensors have also been applied in tactile sensing in recent years. For example, Yuanzhao Wu proposed a tactile sensor based on a GMI sensor [75]. They used the Co-based amorphous wire as the sensitive material and wrapped a copper coil outside the wire to provide a biased magnetic field, as shown in Figure 5a. When there is a force, the flexible layer mixed with magnetic particles deforms, which can be seen in Figure 5b. The change of the magnetic field leads to the change of the impedance of the Co-based amorphous wire. Because the signal is AC, this sensor outputs digital (frequency) signals directly, which is beneficial for subsequent signal processing. This sensor is very sensitive, with a minimum detectable force of 10 µN and a minimum detectable pressure of 0.3 Pa.

However, GMI sensors require an extremely high frequency (100 MHz to GHz) drive signal. The circuit that generates high-frequency signals is also complex. This brings difficulties to the miniaturization and low power consumption of devices. Moreover, GMI sensors need a biased magnetic field when working, so coils or permanent magnets are needed outside GMI materials. In addition, GMI sensors have other drawbacks, such as large residual magnetism, magnetic characteristics susceptible to environmental conditions, and high noise of detection circuit, which limit their future development in tactile sensing technology.

#### 2.1.6. Other Magnetic Sensors

In addition to the above sensors, magnetic sensors also include fluxgate meter, atomic magnetometer, SQUID, etc. Fluxgate meter has high sensitivity and stable performance, which is very suitable for low-frequency weak magnetic field detection. However, the sensitivity of fluxgate meter is proportional to the measurement area of coil, so the volume of high-resolution fluxgate is large, and it is difficult to miniaturize and does not facilitate easy electronic integration [92]. The resolution of the atomic magnetometer can reach several fT without a low-temperature environment. It is widely used in large-scale equipment for outer space exploration, anti-submarine, and anti-mine. However, its volume is even larger than that of fluxgate sensor. Although it can be miniaturized with the help of MEMS technology, it is often difficult to achieve both miniaturization and sensitivity [93,94]. SQUID is the most sensitive magnetic sensor at present, which can detect the magnetic strength on a fT scale or even lower, with wide measurement range and high response frequency. SQUID is also the most mature sensor in biological magnetic field detection, which has important application value in disease testing. However, SQUID can only work at extremely low temperatures, requiring complex devices such as liquid helium or liquid nitrogen cooling. SQUID also needs an excellent magnetic shielding environment when working. Therefore, SQUID is very large in size, high in power consumption, and high in cost [95]. At present, it can only be used in specific application areas, which is not suitable at all in tactile sensing. At present, there are also some studies on hydrogel magnetic sensors and actuators [96,97,98,99]. The hydrogel has excellent biocompatibility when it is used in electronic skin and medical sensors. However, reports of its application in tactile sensors have not appeared. This is also a possible research direction of magnetic tactile sensors in the future.

### 2.2. Source of Magnetic Field

A magnetic sensor itself has no ability of force sensing. It can only sense magnetic field. Therefore, as a medium, the source of magnetic field is necessary for tactile sensing. For example, as mentioned above, there are a large number of studies using permanent magnets as the source of magnetic field. In addition to permanent magnets, magnetic field sources also include permanent magnetic particles, inverse magnetostrictive materials, and coils. They have different characteristics when applied to tactile sensing technology. Next, we will introduce them respectively.

#### 2.2.1. Permanent Magnet

Permanent magnets are the most common source of magnetic fields. There are many kinds of permanent magnet materials, some of which come from nature, such as magnetite. Nowadays, artificial magnets are more widely used, such as ferroalloys doped with Al and Ni elements. Among all kinds of permanent magnets, rare-earth magnets (such as NdFeB and samarium cobalt magnets) have high remanence, large coercivity, and are not easy to demagnetize. They are the most common permanent magnets in tactile sensing.

Permanent magnets are simple to use, and the size and direction of the magnetic field can be flexibly adjusted through the placement of the permanent magnet. In 2019, Muhammad Rosle et al. studied the influence of the number and placement angle of cylindrical permanent magnets on a magnetic angle sensor [48]. They designed eight placement methods, as shown in Figure 6a.

The pressure was applied to the hemispherical flexible body from different angles in the range of 0–90 degrees, and the output was observed to change in the X, Y, and Z axes. The experimental data and fitting curves of eight designs are shown in Figure 6b. For the angle sensor, the more obvious the output difference of different angles, the better the angle resolution. It can be seen that the output changes of design 1, 2, 3, 4, and 8 are not obvious, and it is difficult to distinguish the direction of the force applied on the flexible body. The output of design 5, 6, and 7 change significantly, so these three designs can be used to estimate the direction of the applied force. This study has reference value for the design of magnetic tactile sensors using multiple magnets. The number and placement angle of permanent magnets need to be designed according to actual requirements.

There are lots of designs that use permanent magnets to provide magnetic field [53,54,55,57,62,100,101]. Although permanent magnets are easy to use, they are generally large in size, which is not conducive to the miniaturization of devices. Furthermore, permanent magnets will change the elastic properties of materials, which is not friendly for the flexibility and wearability of devices. In addition, after repeated deformation, the connection position between the permanent magnet and the flexible material is easy to change irreversibly, resulting in the change of initial magnetic field and reducing the durability.

Because of simplicity and high magnetic strength, permanent magnets are the most common magnetic sources in magnetic tactile sensors. In the future, permanent magnets will still be the best choice for tactile sensors without high performance and volume requirements.

#### 2.2.2. Permanent Magnetic Particle

In addition to permanent magnets, permanent magnetic particles can also be the source of magnetic field. Generally, the addition of permanent magnet blocks will reduce the flexibility of devices. Permanent magnetic particles can avoid this problem. The diameter of permanent magnetic particles can be as low as several microns or even several tens of nanometers [60,61]. As long as the volume ratio of flexible body and magnetic particles is well-controlled, the flexibility and magnetism of the sensitive structure can be well-combined. Magnetic particles are also very conducive to the miniaturization of devices, because the number of particles can be configured flexibly according to the size requirements of devices.

There are also many studies on tactile sensing with permanent magnet particles [51,64,65,66,68,75]. Especially in recent years, magnetic skin has been widely studied [102,103]. For instance, Tess Hellebrekers created a magnetic skin with a mixture of silicone and magnetic particles, as shown in Figure 7a [47,104]. A 5 × 5 array was fabricated with a Hall sensor underneath the magnetic skin. The scene of applying this array as a sensing device to play a minesweeping game is shown in Figure 7b. This sensor achieved large-area measurements and could distinguish a minimum force of 30 mN.

There are also a few problems in the application of magnetic particles. After mixing magnetic particles with elastomer, the magnetic field direction of magnetic particles is disordered. Therefore, it is necessary to magnetize the magnetic particles with a strong magnetic field. The magnetization process needs to be completed before the flexible structure is combined with the magnetic sensor, otherwise the strong magnetic field will damage the magnetic sensor. In addition, magnetic particles are prone to aggregation when mixed in the flexible body, resulting in an uneven distribution of the magnetic field. These problems introduce more complexity in the manufacturing of sensors. Despite all this, with the rapid development of flexible electronics, the advantages of easy combination with flexible materials and easy miniaturization means that magnetic particles have great potential in artificial skin and other related applications in the future.

#### 2.2.3. Inverse Magnetostrictive Material

In addition to permanent magnetic materials, inverse magnetostrictive materials can also be applied to provide magnetic fields. For example, when a Fe-Ga alloy (Galfenol) deforms due to external force, the magnetic strength around the material will change. The magnitude of the external force can be converted by detecting this change with a magnetic sensor.

Michael Marana designed a flow and tactile sensor with Galfenol as the magnetic field source [59]. The change of magnetic field was detected by a GMR sensor placed around the Galfenol beam. The structure of the sensor is shown in Figure 8a. Finally, a sensitivity of 0.51 mV/mm for the end displacement of the cantilever beam was achieved. In order to meet the requirements of tactile sensors for small size and flexibility, Ling Weng et al. manufactured a tactile sensor on a flexible printed circuit board. The structure is shown in Figure 8b. They used three Galfenol beams as cantilever beams. One end of the cantilever beam was fixed, and the other end bent with force, resulting in a change of magnetic field. They successively used commercial Hall and TMR sensors to detect the change [49,72,105,106]. The sensitivity of this device is 126 mV/N in the range of 0–3 N.

However, the inverse magnetostriction effect requires a permanent magnet to provide a bias magnetic field. Furthermore, for magnetostrictive elements, the length of the cantilevers needs to be large enough to provide sufficient sensitivity. In the future, the miniaturization of reverse magnetostrictive devices is the most important problem to be solved when they are used in tactile sensing technology.

#### 2.2.4. Coil

Coil is one of the most primitive and simplest magnetic field generating devices. Its principle is Faraday’s law of electromagnetic induction. Wattanasarn et al. proposed a three-dimensional magnetic tactile sensor with coils embedded in a flexible body. The structure diagram is shown in Figure 9 [77]. There are two layers of coils in total. The upper layer are detection coils, while the lower layer are excitation coils. The coils in each layer are arranged in two rows and two columns. An elastic pillar and a spacer layer are sandwiched between the two layers. When the external force is applied to the flexible body, the distance between two layers changes, thereby changing the output of detection coils. There are some similar studies with coils as the source of magnetic field [50,107].

The method of providing the magnetic field with coils is very simple, and the strength of the magnetic field can be flexibly adjusted. This is advantageous to improve the sensing range of the tactile sensor. However, there are also some problems of coils in tactile sensing. Firstly, to improve the performance, the size of the coils is often large, and an iron core is often needed. Therefore, the volume of the device is generally large. Secondly, compared with the permanent magnet, the coil needs continuous power supply when providing the magnetic field, so the power consumption of the device is very large. These problems hinder the application of coils in tactile sensing. Recently, micro-excitation coils have also been studied [108,109]. The application of micro-excitation coils to the magnetic tactile sensors can greatly reduce the size of tactile devices, which is worth further study in the future.

### 2.3. Structure of Sensitive Body

The structures of flexible sensitive bodies also play a great role in magnetic tactile sensors. The common structures are convex structure, film structure, and ciliary structure. Their characteristics are introduced respectively below.

#### 2.3.1. Convex Structure

As previously mentioned, the convex structure is the most common tactile sensing structure. As shown in Figure 10 [42,43,53,81], the shapes of convex structures include hemispherical, pyramid, cylindrical, etc. Compared with the film structure, the convex structure has better detection ability for the tangential force in addition to the normal force. Therefore, it is more suitable for the detection of multi-dimensional tactile force. In addition, because a part of the convex structure is higher than the plane, it can detect more complex and diverse surfaces, including bulges, planes, and even depressions.

Some studies added air gaps in convex structures. Air gaps are very effective to reduce the Young’s modulus of the sensitive structure, and help sensors get higher sensitivities. However, air gaps also make sensors easier to be saturated, resulting in a reduction in detection ranges. In practical applications, whether to use air gaps or not should be judged according to needs.

There are some problems in the application of convex structures. When many convex structures are combined to form a large area array, the surface of the array will be uneven, which cannot meet the high requirements for surface continuity and consistency of tactile skin. For example, for the same normal force, the signal output at gaps and vertices of convex structures is different. This makes it difficult to characterize objects with sharp or other complex structures.

#### 2.3.2. Film Structure

Film structures are more suitable for large-area tactile sensing when combined with a sensor array. For example, Tito Tomo et al. proposed a tactile sensor with a simple film structure in 2016, as shown in Figure 11a [44]. There is a permanent magnet in the center of the film. This sensor can distinguish a minimum force of 10 mN. Later, they miniaturized the sensing structure and made it into an array. Then, they put the array on the finger muscle belly of a robot [110]. In 2017, they further applied this sensor array to robot fingertips, which brought the spatial integration of magnetic tactile sensors to a new level [45]. In addition to adding permanent magnets to the films, more studies used the method of adding magnet particles into the films, like the film shown in Figure 11b [67]. This is advantageous for improving the flexibility and magnetic consistency of the device.

Decoupling the signals of normal force and tangential force is one of the difficulties in tactile sensing technology. At present, it is necessary to design complex sensing structures or use complex mathematical models to realize decoupling. In addition, decoupling and super-resolution detection cannot be achieved at the same time. To solve these problems, professor Shen Yajing cooperated with professor Pan Jia and proposed a new tactile sensor based on magnetic films [51]. The stereo view of the sensor is shown in Figure 12a. The magnetic films they used were magnetized into Halbach arrays, as shown in Figure 12b. Red and blue dots represent opposite magnetization directions. An important advantage of this structure is that it can enhance the magnetic strength on one side of the film, which can improve the sensitivity of the sensor. When applying this film to the tactile sensor, they found an important feature where the normal force and tangential force are naturally decoupled. As shown in Figure 12c,d, the magnetic strength B is only related to the normal force, not to the tangential force. Further, the magnetic ratio R_B_ (equal to Bx/Bz, where Bx and Bz are the magnetic strength along the x and z directions, respectively) is related to the tangential force only. In addition, since the distribution of magnetic field in space is continuous, super-resolution can be achieved by an algorithm. They used neural network algorithm for signal processing, which greatly improved the detection resolution of the sensor. Finally, the spatial resolution of this sensor is 60 times smaller than the sensor spacing.

Film structure is very suitable for large-area tactile sensing. Based on this, magnetic skin is a promising research direction of artificial skin. Large-area tactile sensing requires a large number of magnetic sensors to form an array, which has high requirements for the consistency and miniaturization of magnetic sensors. The fabrication of fully flexible magnetic sensors is also a problem that must be solved. Moreover, the increase in the number of sensors also means that the signal reading and anti-interference circuits become more complex, which reduces the portability and wearability of the tactile sensor.

#### 2.3.3. Ciliary Structure

The ciliary structure is one of the most sensitive structures known in nature. It is often seen in the bodies of natural creatures, such as the side lines of fish, the hairs on spider feet, and the antennae of mosquitoes. The biological ciliary structure is mainly composed of two parts. One is a hairy structure that will deform under force, and the other is nerve cells that can feel this deformation. With the bionics of these ciliary structures, the application of ciliary structure is of great help to improve the detection ability of tactile sensors [111,112,113,114]. In recent years, researchers have combined ciliary structures with various sensors, including magnetic sensors.

Since 2014, Alfadhel Ahmed and Pedro Ribeiro have successively carried out tactile studies with ciliary structures [60,61,65,66,73,74,115,116]. Alfadhel Ahmed firstly produced cilia made of PDMS and Fe nanowires. The structure and working principle of the sensor are shown in Figure 13a. A GMR film was used to detect the change of magnetic signal, so that the change of external force could be detected. In order to make Fe nanowires, they first prepared porous alumina with two anodic oxidation processes, and then produced iron nanowires on porous alumina with electroplating process. Finally, with the help of a PMMA mold fabricated by laser drilling, the cilia were made on the top of the GMR film by the demoulding method, as shown in Figure 13b.

Based on the same principle, Pedro Ribeiro conducted similar research [65]. PDMS was also used as the flexible material, but the source of the magnetic field was replaced by NdFeB magnetic particles. The magnetic strength of NdFeB particles is greater than that of Fe nanowires, so the sensitivity of the sensor is greater. The performance of this sensor is more excellent, and the minimum force of 333 µN can be distinguished. Furthermore, with a GMR sensor, in order to further improve the resolution of the device, Alfadhel Ahmed tested the performance of a single cilium [61]. The single cilium is able to distinguish a smaller force, which is as low as 31 µN.

Although ciliary structures have high sensitivity and are able to detect much smaller forces, there are still many problems to be solved. Firstly, the detection range is small, which is limited by the small Young’s modulus of cilia. Secondly, the content of magnetic particles in cilia is very low, so the magnetic strength around cilia is similar or even smaller than that of the geomagnetic field. Therefore, the geomagnetic field cannot be simply treated as environmental noise. Generally, additional shielding devices or complex circuit processing are required. Thirdly, after long-term use, the cilia will be irreversibly tilted or damaged, which brings great challenges to the durability of the sensor. In addition, compared with normal force, cilia are more suitable for measuring tangential force. When the normal force acts on the cilia, the direction of cilia bending is random, and the linearity of the signal is poor. This leads to the complexity of signal extraction and processing.

## 3. Application

Magnetic tactile sensing technology is still immature, but it has been applied in some specific application areas. There are a number of studies that have tried various cutting-edge applications, mainly including robot precision grasping, texture characterization, flow velocity measurement, and medical treatment. These applications are summarized below.

### 3.1. Robot Precision Grasping

Tactile sensors can provide feedback signals for robots, and help robots realize precision operations such as grabbing fragile objects. In recent years, there have been many manipulators using magnetic sensors as tactile feedback elements [40,45,46,69,81,110], as shown in Figure 14. For example, Alireza Mohammadi integrated multiple magnetic sensors with convex structures into a manipulator, as shown in Figure 14a [81]. When the manipulator grabs objects, the convex structures will squeeze. The permanent magnets in the convex structure are closer to the Hall sensor, thus generating the output signal. Such sensors can be installed in different positions of the manipulator according to requirements, such as fingertips, finger pulps, and palms. Different positions have different requirements for sensor structures. The fingertip generally requires a sensor with convex structures, while the finger pulps and palms generally require a sensor with film structures.

There are three main problems when integrating magnetic tactile sensors into manipulators. Firstly, magnetic sensors do not have the ability to sense force directly. They must be combined with magnetic materials to work. Therefore, the volume of magnetic tactile sensors is difficult to reduce, resulting in low sensing spatial resolution of manipulators. Secondly, for tactile manipulators, the magnetic sensors and actuators are all needed, and a large number of connecting wires are required, which make the manipulators very cumbersome. Thirdly, the mechanical structures and actuators of manipulators often contain ferromagnetic materials. The magnetic field produced by these devices is often difficult to shield and eliminate. When manipulators grab objects with ferromagnetic materials, there will also be interference signals.

In order to solve these problems, several studies have been carried out in recent years. Inspired by human tactile sensing and synaptic transmitter, Sunjong Oh et al. designed a remote tactile sensing system with a GMR sensor. Instead of using a large number of electric wires, they used some air tubes on a manipulator [63]. At the front end of fingertips, there are only flexible films and air gaps, as shown in the enlarged figure (fingertip) in Figure 15a. The sensors and signal processing circuits are far away from fingertips and connect with the fingertips through air tubes, as shown in Figure 15b. This is conducive to the miniaturization of the manipulator. Moreover, because magnetic sensors are far away from the manipulator, the ferromagnetic materials on the manipulator or the ferromagnetic objects being grabbed will not interfere with the magnetic sensors. They installed the sensor into a manipulator and tested its performance [54]. The resolution of this sensor is as low as 300 µN. The air tubes used in this study have important reference value for the future development of magnetic tactile sensors with high density and low noise.

### 3.2. Texture Characterization

Texture characterization can be applied in object state recognition, safety human–machine interaction, food safety, and other application areas. Current methods of texture characterization are mainly based on two principles: optics [117] and acoustics [118]. However, these methods often have low portability and high power consumption, so they are difficult to be integrated on robot platforms. The method based on the piezoelectric principle has also been studied, which is very promising. Topographical scanning capable of resolving features with 25 µm thickness have been reported [119].

In recent years, magnetic sensors have also been applied in texture characterization. Texture characterization has high requirements for the resolution of sensors, so the ciliary structure is often used. For example, Pedro Ribeiro developed a texture sensor with TMR nano-films and nine cilia, as shown in the top right of Figure 16a [120,121]. Compared with their previous studies [61], the magnetic cilia they used this time had a higher mass content (65%) of magnetic particles. The test diagram is shown in Figure 16b. The height of the texture is 20 µm or 50 µm. The blue line and red line are the test results of the 1 × 3 (top) and 3 × 3 (bottom) cilia array, respectively. It can be seen that with a 1 × 3 cilia array, this sensor successfully recognizes the texture of a height of 20 µm. This is the lowest known texture height that can be detected with a tactile sensor. Benefiting from its powerful ability, this sensor was also used to distinguish the rotten state of fruits [71,122], as shown in the left of Figure 16a. When cilia crossed the surface of different fruits, the rotten state of fruits could be recognized according to the different signal of the sensor.

### 3.3. Flow Velocity Measurement

In nature, a lot of organisms can sense the changes of external flow with the help of ciliary structures. For example, spiders can perceive changes in air flow by the ciliary structures on their legs, and fish can perceive the speed of water flow by their side lines. Similar to force, the flow of air or water will also lead to the bending of magnetic cilia. The flow velocity can be obtained by detecting the change of stray magnetic field with magnetic sensors.

For example, Ahmed Alfadhel et al. also applied their ciliary sensors mentioned above to the measurement of flow velocity [73]. They tested air flow and water flow separately. The experimental setup is shown in Figure 17. The sensitivity of the sensor to air flow and water flow is 24 mΩ/(mm/s) and 0.9 Ω/(mm/s), respectively. The minimum air flow of 0.56 mm/s and water flow of 15 µm/s can be distinguished. The power consumption of the sensor is as low as 36.1 µW, which shows that this sensor has great application potential in low-power flow velocity detection.

For the detection of water flow velocity, the waterproofing of the device is very important. For piezoresistive or piezoelectric sensors, because current flows in the sensitive structure exposed to water, it can be easily damaged. Magnetic sensors can solve this problem. There is no current in the magnetic cilia, which is completely independent and separated from the magnetic sensor. In contrast, the magnetic sensor only needs to be completely sealed to be waterproof. In other words, the non-contact measurement ability of the magnetic sensor makes the waterproofing of the device very simple.

### 3.4. Medical Treatment

In medical treatments, several experimental explorations with magnetic tactile sensors have been carried out [123]. The most common way is to extract physiological signals with the sensors. For example, Sunjong Oh et al. tested the performance of a magnetic tactile sensor in reading pulse signals [63]. The experimental scenario and results are shown in Figure 18a,b. The blue and red curves are the pulse signals of a volunteer before and after exercise, respectively. It can be seen that the amplitude of the red curve is larger and the frequency is higher. The difference between the two signals is very obvious, and the heart rate can be extracted from the signal.

Another attempt of magnetic tactile sensors in medical treatment is the rehabilitation of people with visual impairment. Alfadhel Ahmed et al. also designed a magnetic tactile sensor that can read Braille with a GMR sensor [116]. The 2 × 3 Braille dots (corresponding to 26 English letters) and test diagram are shown in Figure 18c. They used four tactile sensing elements to read Braille dots. Among them, two elements (R1 and R2 in Figure 18c) were used as sensing elements, and the other two were reference elements to reduce environmental noise. As shown in Figure 18d, when the cilia at R1 and R2 contact Braille dots, the sensor will output different voltages according to whether there are dots, thus Braille recognition is realized.

High-performance tactile sensors will play an important role in medical treatment in the future. The ciliary structure greatly improves the sensitivity of magnetic tactile sensors, which makes these sensors suitable for application in specific areas such as minimally invasive surgery. However, medical treatment has extremely high requirements for the reliability of devices. Due to the problems of ciliary structures mentioned in Section 2.3.3, such sensors have not yet been commercialized in medical treatment.

## 4. Research Difficulties and Future Directions

Although a large number of studies have been done, magnetic tactile sensors have not been widely used at present. There is a long way from commercial application. The main reasons are as follows:A magnetic sensor itself has no ability to sense force, so it must work with additional ferromagnetic materials and flexible structures to realize tactile sensing. Therefore, how to reduce the size of the whole device is a key problem in the application of magnetic tactile sensors.Magnetic tactile sensors are sensitive to ferromagnetic objects in the environment. Highly sensitive magnetic sensors are also very sensitive to the geomagnetic field, which is almost everywhere. Therefore, when magnetic sensors are applied to tactile sensing, there is a lot of magnetic noise, which is difficult to shield.Large-area sensing is an inevitable research direction of tactile sensing technology, so research on the sensor array is very important. However, there are many difficulties in the research of magnetic sensor array, including: how to improve the consistency of magnetic sensors in the array, how to improve the spatial resolution of the array (reach or exceed the spatial resolution of human skin), how to simplify the signal reading circuit of the large-area array while ensuring the reading speed, etc.Tactile sensors emphasize high requirements for the flexibility of devices. At present, the fabrication of magnetic sensors is mostly based on semiconductor technology and nano-film deposition technology. In most cases, these technologies are implemented on hard substrates such as silicon wafers. The process of producing magnetic sensors on flexible substrates is immature, and the performance of fully flexible sensors will be greatly reduced or even damaged during bending and stretching.

These problems bring great challenges to researchers, and the prospects are not optimistic in the short term.

The future development of applied technology generally depends on the breakthrough of basic research. For magnetic tactile sensors, the improvement of performance greatly depends on the breakthrough of magnetic sensors in materials and processes. For example, TMR sensors, which are currently being widely studied, have great application potential in tactile sensing. By finding different material systems, or proposing more efficient and low-cost processes, or with the help of chopper technology or MEMS modulation methods, it is possible to manufacture TMR devices with less low-frequency noise and higher sensitivity. This is of great significance for the development of magnetic tactile sensing technology. In the future, magnetic tactile sensing technology can expand its scope of application. Especially in medical and health care, it is worth trying to install sensors on surgical instruments such as endoscopes to judge small changes in organs or determine the location of tumors. Magnetic tactile sensors also have the potential to play a huge role in tactile feedback of telemedicine robots. In addition, compared with other tactile sensors, non-contact measurement is a major advantage of magnetic tactile sensors. It is possible to realize non-invasive detection of medical information by combining magnetic tactile sensors with magnetic markers, such as magnetic nanoparticles. Last but not least, it is difficult to complete complex tasks by using only tactile sensors. With the help of multi-sensor information fusion technology, combining tactile sensors with other sensors (such as visual sensors) can make robots acquire more comprehensive and advanced capabilities. This is an important way for the widespread use of magnetic tactile sensors in the future.

## 5. Conclusions

This paper introduces the classification, development, and application of magnetic tactile sensors in detail. Many types of magnetic sensors with different characteristics have been applied to tactile sensing technology up to now. Among them, the most common and practical types are Hall and GMR sensors. Due to the obvious shortcomings in size, power consumption, or noise, other types of magnetic sensors are not the first choice for tactile sensing. With the development of magnetic sensors, the detection of the minimum force on a millinewton scale has now been realized successfully. The most common source of magnetic field in current research is the permanent magnet. Magnetic particles are easy to combine with flexible materials and have great potential in the future development of flexible electrons. For the structures used by magnetic tactile sensors, the film is the preferred structure for large-area perception, while the ciliary structure is more suitable for high-precision perception of tactile force. 

Up to now, magnetic tactile sensors have not been widely used. In order to narrow the gap between academia and industry, researchers should pay more attention to the basic research of magnetic sensors in materials and processes, and improve the spatial resolution, robustness, and array consistency of magnetic tactile sensors. In the event that the advantages of high sensitivity, three-dimensional detection, and non-contact measurement could be fully exploited, the application scope of magnetic tactile sensors in intelligent robots and modern medicine can be further expanded in the future.

## Figures and Tables

**Figure 1 biosensors-12-01054-f001:**
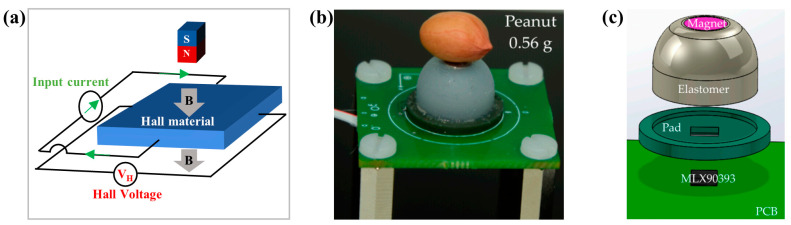
(**a**) Illustration of the Hall effect; (**b**) one of the typical tactile sensors with MLX90393 (designed by Hongbo Wang). Reprinted with permission from Ref. [42]. Copyright 2016, MDPI; (**c**) structure diagram of the tactile sensor with MLX90393. Reprinted with permission from Ref. [42]. Copyright 2016, MDPI.

**Figure 2 biosensors-12-01054-f002:**
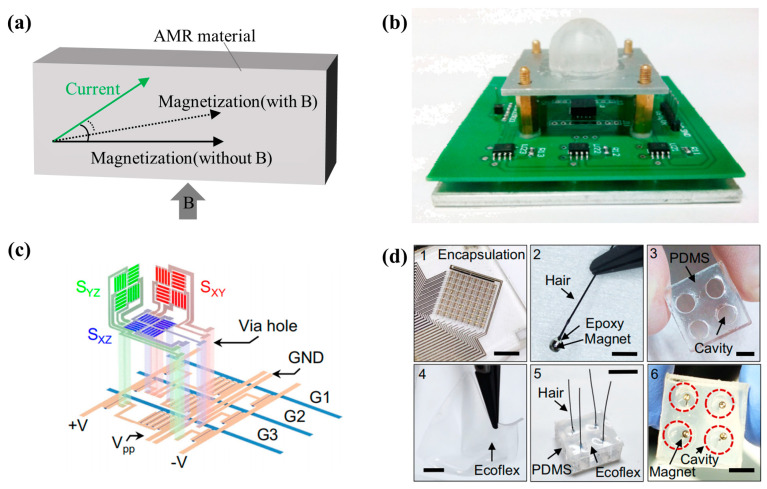
(**a**) Illustration of AMR effect; (**b**) Tactile sensor with a commercial 3-axis AMR sensor (designed by Ping Yu). Reprinted with permission from Ref. [53]. Copyright 2014, Springer Nature; (**c**) Three-dimensional sensitive structure of the tactile sensor with AMR film (designed by Christian Becker). Reprinted with permission from Ref. [55]. Copyright 2022, Springer Nature; (**d**) Manufacturing process of the sensor. 1, The three-dimensional AMR sensor array is encapsulated by epoxy. Scale bar, 4 mm. 2, A small magnet is adhered to a hair and embedded in epoxy. Scale bar, 2 mm. 3, A spacer PDMS (polydimethylsiloxane) film with cavities is fabricated. Scale bar, 5 mm. 4, Ultra-flexible Ecoflex film is casted. Scale bar, 2 mm. 5, Magnetic hair, PDMS spacer and Ecoflex skin are assembled. Scale bar, 15 mm. 6, Bottom view of the skin layer showing that the magnets are hold within the cavities of the PDMS spacer. Scale bar, 5 mm. Reprinted with permission from Ref. [55]. Copyright 2022, Springer Nature.

**Figure 3 biosensors-12-01054-f003:**
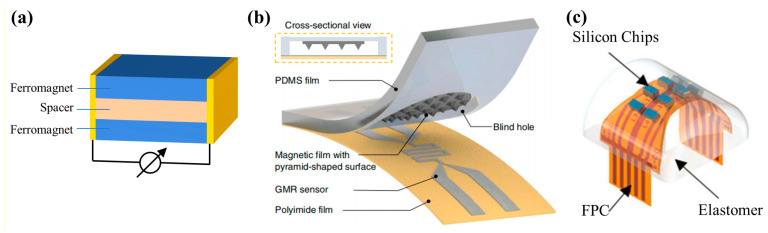
(**a**) Illustration of the spin valve structure of GMR sensors; (**b**) Thin-film magnetic tactile skin with GMR multilayer films (designed by Jin Ge). Reprinted with permission from Ref. [68]. Copyright 2019, Springer Nature; (**c**) Flexible-rigid hybrid tactile sensor (designed by Miguel Neto). Reprinted with permission from Ref. [69]. Copyright 2021, MDPI.

**Figure 4 biosensors-12-01054-f004:**
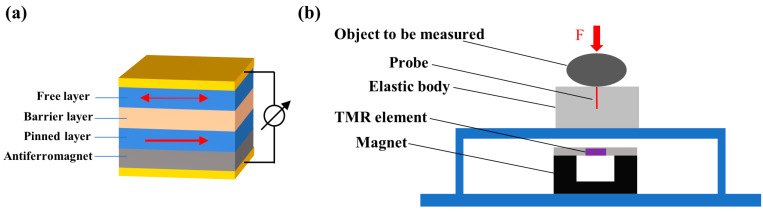
(**a**) Illustration of the structure of TMR sensors; (**b**) Structural diagram of the tactile sensor with a commercial TMR sensor.

**Figure 5 biosensors-12-01054-f005:**
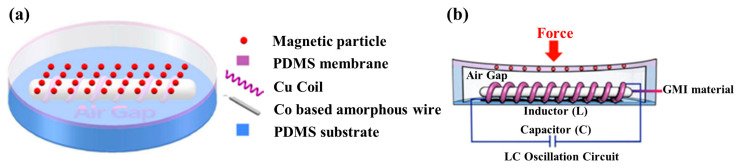
(**a**) Structure diagram of the tactile sensor with a GMI sensor (designed by Yuanzhao Wu). Reprinted with permission from Ref. [75]. Copyright 2018, The American Association for the Advancement of Science; (**b**) Working state when a force acts on the sensor. Reprinted with permission from Ref. [75]. Copyright 2018, The American Association for the Advancement of Science.

**Figure 6 biosensors-12-01054-f006:**
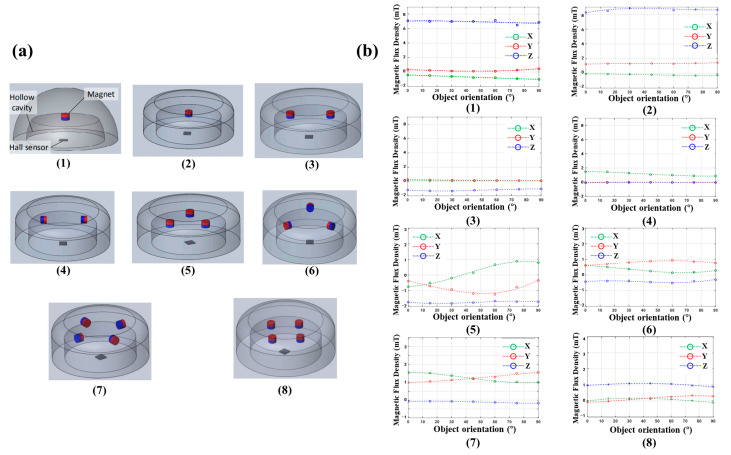
(**a**) Eight placement methods of permanent magnets. Reprinted with permission from Ref. [48]. Copyright 2019, MDPI; (**b**) Experimental data and fitting curves of 8 designs. The green, red and blue circles are the magnetic flux densities of the X, Y and Z axes measured by the sensor. The dotted green, red and blue lines are the fitting curves. Reprinted with permission from Ref. [48]. Copyright 2019, MDPI.

**Figure 7 biosensors-12-01054-f007:**
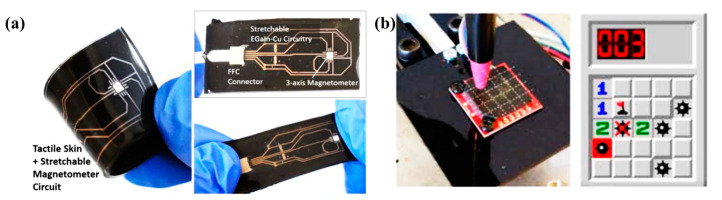
(**a**) Magnetic skin using a mixture of silicone and magnetic particles created by Tess Hellebrekers. Reprinted with permission from Ref. [47]. Copyright 2019, Wiley; (**b**) The scene of applying this array as a sensing device to play a minesweeping game. Reprinted with permission from Ref. [47]. Copyright 2019, Wiley.

**Figure 8 biosensors-12-01054-f008:**
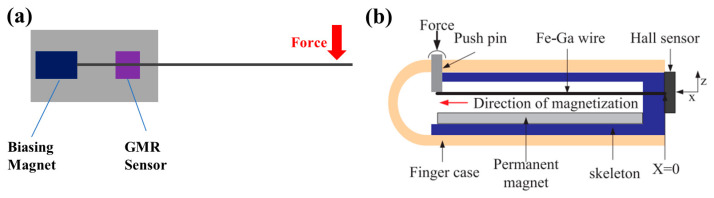
(**a**) Structure diagram of the sensor with a Galfenol beam (designed by Michael Marana); (**b**) Structure diagram of the sensor with three Galfenol beams (designed by Ling Weng). Reprinted with permission from Ref. [49]. Copyright 2020, Elsevier.

**Figure 9 biosensors-12-01054-f009:**
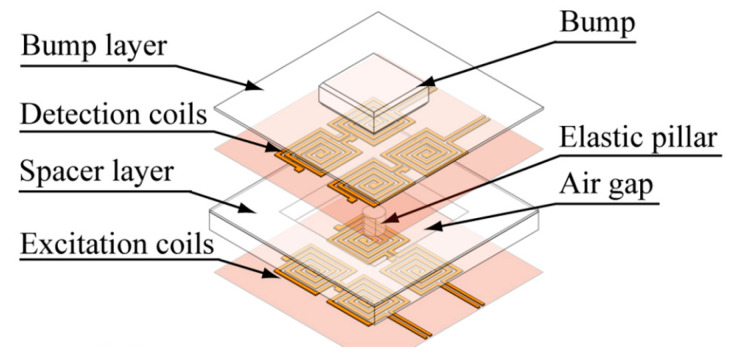
The structure diagram of the tactile sensor with coils (designed by Wattanasarn). Reprinted with permission from Ref. [77]. Copyright 2012, IEEE.

**Figure 10 biosensors-12-01054-f010:**
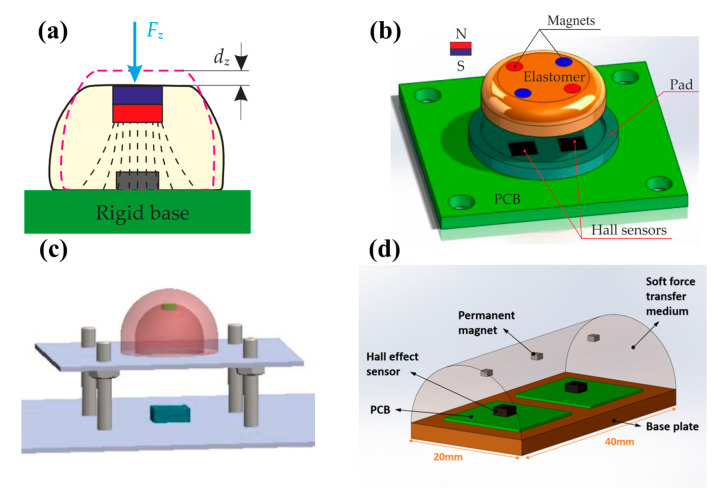
All kinds of shapes of convex structures used in magnetic tactile sensors. (**a**) Magnetic tactile sensor with a convex structure (designed by Hongbo Wang). Reprinted with permission from Ref. [42]. Copyright 2016, MDPI. (**b**) Magnetic tactile sensor with a convex structure (designed by Hongbo Wang). Reprinted with permission from Ref. [43]. Copyright 2016, Elsevier. (**c**) Magnetic tactile sensor with a convex structure (designed by Ping Yu). Reprinted with permission from Ref. [53]. Copyright 2014, Springer Nature. (**d**) Magnetic tactile sensor with a convex structure (designed by Alireza Mohammadi). Reprinted with permission from Ref. [81]. Copyright 2019, MDPI.

**Figure 11 biosensors-12-01054-f011:**
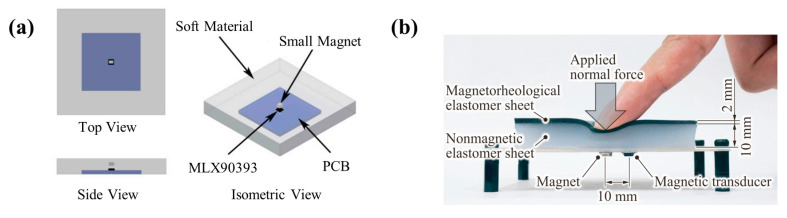
(**a**) Tactile sensor using a film with a permanent magnet in the center. Reprinted with permission from Ref. [44]. Copyright 2016, MDPI; (**b**) Tactile sensor using a film with magnet particles. Reprinted with permission from Ref. [67]. Copyright 2018, MDPI.

**Figure 12 biosensors-12-01054-f012:**
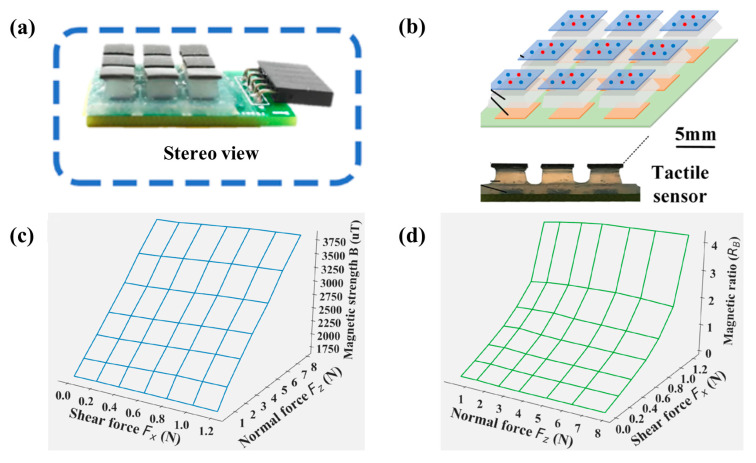
(**a**) Stereo view of the soft tactile sensor with a flat surface in a 3-by-3 array. Reprinted with permission from Ref. [51]. Copyright 2021, The American Association for the Advancement of Science; (**b**) Structure of the sensor with magnetic films which are magnetized into Halbach arrays. Reprinted with permission from Ref. [51]. Copyright 2021, The American Association for the Advancement of Science; (**c**) The magnetic strength B under both shear and normal forces. Reprinted with permission from Ref. [51]. Copyright 2021, The American Association for the Advancement of Science; (**d**) The magnetic ratio R_B_ under both shear and normal forces. Reprinted with permission from Ref. [51]. Copyright 2021, The American Association for the Advancement of Science.

**Figure 13 biosensors-12-01054-f013:**
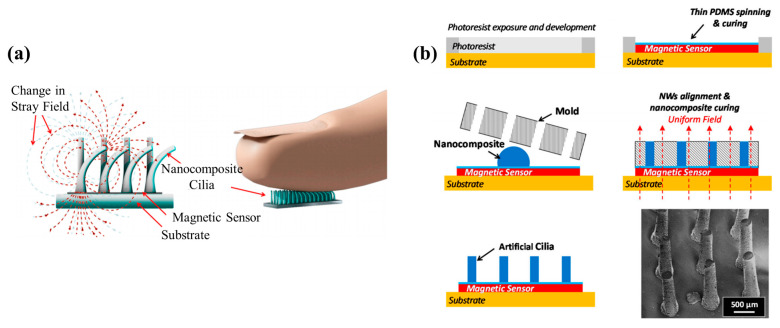
(**a**) Structure and working principle of the sensor with ciliary structure (designed by Alfadhel Ahmed). Reprinted with permission from Ref. [60]. Copyright 2016, MDPI; (**b**) Manufacturing process of the sensor. Reprinted with permission from Ref. [60]. Copyright 2016, MDPI.

**Figure 14 biosensors-12-01054-f014:**
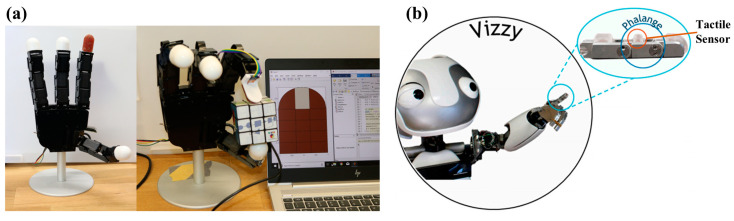
(**a**) Manipulator with magnetic tactile sensors designed by Alireza Mohammadi. Reprinted with permission from Ref. [81]. Copyright 2019, MDPI; (**b**) Manipulator with magnetic tactile sensors designed by Miguel Neto. Reprinted with permission from Ref. [69]. Copyright 2021, MDPI.

**Figure 15 biosensors-12-01054-f015:**
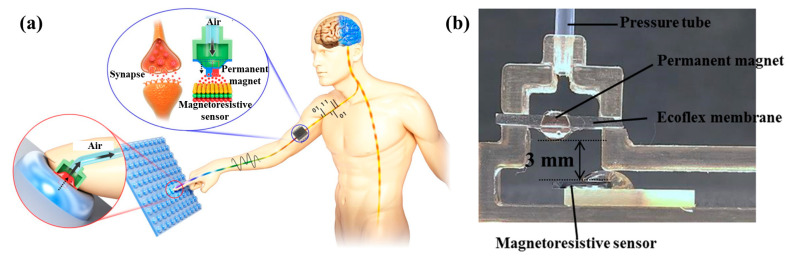
(**a**) Schematic illustration of remote tactile sensing system with magnetic synapse inspired by human tactile sensing and synaptic transmitter. Reprinted with permission from Ref. [63]. Copyright 2017, Springer Nature; (**b**) Cross-sectional image of magnetic synapse with air tube connected. Reprinted with permission from Ref. [63]. Copyright 2017, Springer Nature.

**Figure 16 biosensors-12-01054-f016:**
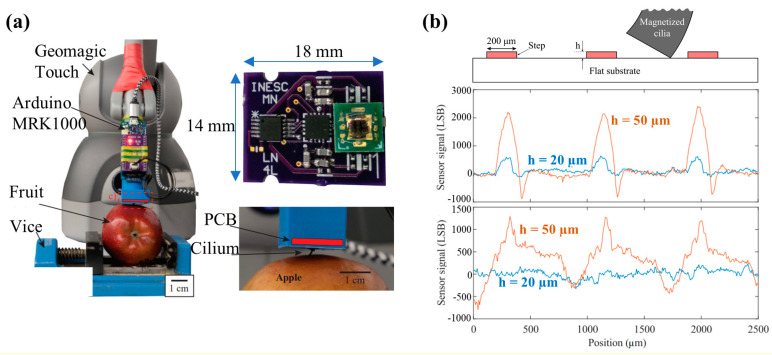
(**a**) Photograph of the tactile sensor with 9 cilia (designed by Pedro Ribeiro) and the test scenario when it is used to distinguish the rotten state of fruits. Reprinted with permission from Ref. [71]. Copyright 2020, IEEE; (**b**) Test diagram of texture characterization and the test results of 1 × 3 (top) and 3 × 3 (bottom) cilia array. Reprinted with permission from Ref. [120]. Copyright 2020, IEEE.

**Figure 17 biosensors-12-01054-f017:**
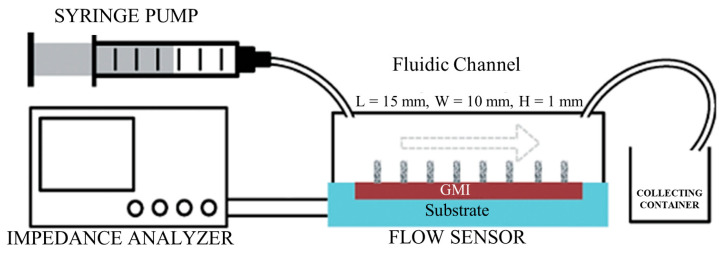
Experimental setup of the flow sensor designed by Ahmed Alfadhel. Reprinted with permission from Ref. [73]. Copyright 2014, ROYAL SOCIETY OF CHEMISTRY.

**Figure 18 biosensors-12-01054-f018:**
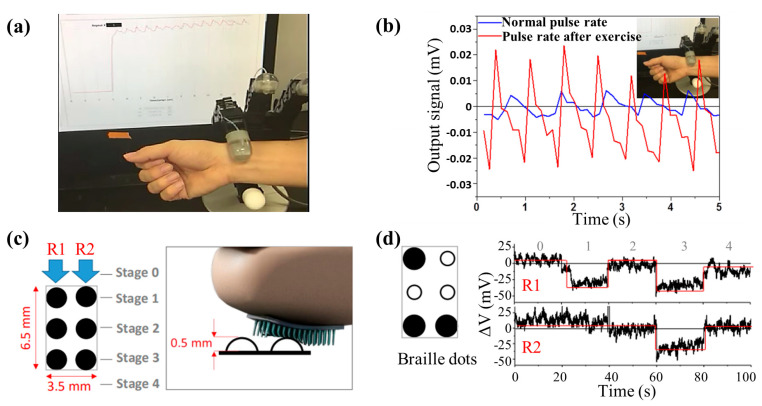
(**a**) Experimental scenario when the sensor is used to read pulse signal. Reprinted with permission from Ref. [63]. Copyright 2017, Springer Nature; (**b**) The pulse signals of a volunteer before and after exercise. Reprinted with permission from Ref. [63]. Copyright 2017, Springer Nature; (**c**) 2 × 3 Braille dots (corresponding to 26 English letters) and the test diagram. Reprinted with permission from Ref. [116]. Copyright 2016, IEEE; (**d**) Test results when the sensor is used to distinguish Braille dots. Reprinted with permission from Ref. [116]. Copyright 2016, IEEE.

**Table 1 biosensors-12-01054-t001:** Comparison of magnetic tactile sensors of different types.

Sensor Type	Year	Author/Ref	Magnetic Field Source	Sensitivity	Resolution	Range
Hall	2006	Eduardo [37]	magnet	-	94 mN	-
	2013	Ledermann [38]	magnet	6.88 LSB/N	150 mN	12 N
	2013	Jun-ichiro YUJI [39]	magnet	0.38 mV/N	-	0–50 N
	2014	Sina Youssefuan [40]	magnets	0.5 N/(mm/mm)	-	-
	2015	Lorenzo Jamone [41]	magnet	-	10 mN	3 N
	2016	Hongbo Wang [42]	magnet	85–298 Gauss/N (x, y) 42–83 Gauss/N (z)	0.71 mN (x, y) 1.42 mN (z)	3.4 N
	2016	Hongbo Wang [43]	magnet	-	1.5 mN	20 N
	2016	Tito Tomo [44]	magnet	-	10 mN	0.7–14.5 N
	2017	Tito Tomo [45]	magnet	-	20 mN	6.3 N
	2018	Anany Dwivedi [46]	magnet	-	5 mN	1.5 N (x, y) 1.1 N (z)
	2019	Tess Hellebrekers [47]	particle	-	30 mN	0.03–1.9 N
	2019	Muhammad Rosle [48]	magnet	0.08 mT/N	-	6.5 N
	2020	Ling Weng [49]	magnetostriction	126 mV/N	-	0–3 N
	2020	Alexis C. Holgado [50]	coil + magnet	Adjustable	-	-
	2021	Youcan Yan [51]	particle	0.1–0.27 kPa^−1^ (x) 0.01 kPa^−1^ (z)	-	0–16 kPa (x) 0–120 kPa (z)
	2022	Muhammad Rehan [52]	magnet	16 mV/N	5 mN	0–20 N
AMR	2014	Ping Yu [53]	magnet	78 mV/N (x, y) 58 mV/N (z)	10 mN	0–4 N (x, y) 0–20 N (z)
	2018	Sang-Hun Kim [54]	magnet	0.016 mV/kPa	300 µN	6 Pa–400 kPa
	2022	Christian Becker [55]	magnet	-	-	-
GMR	2006	Kathleen Hale [56]	magnetostriction	-	-	-
	2010	Masanori Goka [57]	magnet	-	60 mN	−40–40 N
	2012	Hiroyuki Nakamoto [58]	magnet	-	60 mN	−40–40 N
	2012	Marana [59]	magnetostriction	0.51 mV/mm	-	0–65 mm
	2016	Ahmed Alfadhel [60]	nanowire	46 Ω/mN	1.3 mN	15 mN
	2016	Ahmed Alfadhel [61]	nanowire	1.6 Ω/mN	31 µN	1 mN
	2016	Hiroyuki Nakamoto [62]	magnet	-	60 mN	−40–40 N
	2017	Sunjong Oh [63]	magnet	0.126 mV/kPa	6 Pa	6 Pa–400 kPa
	2017	Jung Jin Park [64]	nanowire	1–4 mΩ/kPa	40 µm	-
	2017	Pedro Ribeiro [65]	particle	-	630 µN	0–26 mN
	2017	Pedro Ribeiro [66]	particle	9.62 mN/mV	333 µN	7.8 mN
	2018	Takumi Kawasetsu [67]	magnetorheological elastomer	161 mV/N 12.7 mV/kPa	10 mN	2.5 N
	2019	Jin Ge [68]	particle	-	-	-
	2021	Miguel Neto [69]	magnet	-	-	0.1–5 N
TMR	2017	Zhang dongfang [70]	magnet	-	-	-
	2020	Pedro Ribeiro [71]	particle	-	-	-
	2022	Huiwen Yang [72]	magnetostriction	323 mV/N	50 mN	0–4 N
GMI	2014	Ahmed Alfadhel [73]	nanowire	Air: 24 mΩ/(mm/s) Water: 0.9 Ω/(mm/s)	Air: 0.56 mm/s Water: 15 µm/s	Air: 190 mm/s Water: 7.8 mm/s
	2015	Ahmed Alfadhel [74]	nanowire	0.55 µV/kPa (in 0–32 kPa)	2.7 kPa	0–120 kPa
	2018	Yuanzhao Wu [75]	particle	120 N^−1^ 4.4 kPa^−1^	10 µN 0.3 Pa	0–1 kPa
Coil	2009	Satoru Takenawa [76]	magnet	-	60 mN	−40–40 N
	2012	S. Wattanasarn [77]	coil	0.68 mV/N	-	-
	2017	Lili Wan [78]	magnetostriction	2 mV/(kA/m)	-	6 N

“-” represents there is no relevant information in the article.

## Data Availability

Not applicable.

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
