# Peer review of "Recent Progress of Biomimetic Tactile Sensing Technology Based on Magnetic Sensors"

_biosensors, 2022, doi:10.3390/bios12111054_

Round 1

Reviewer 1 Report

The paper represents a useful overview of recent development in magnetic tactile sensing technology. I have only a few notes on factual statements in the paper. Moreover, while the text is mostly written in good English, there are multiple typos and stylistic constructs that should be reconsidered.

The factual notes/objections:

Line 356-358: The diameter of permanent magnetic particles can be as low as several microns or even several nanometers [61,62]. – I strongly recommend rewording to:

“… even several tens of nanometers [61,62].”

(Nanoparticles below 10 nm diameter typically display superparamagnetic behavior, meaning it can no longer be considered a permanent magnetic particle. However, such particles can be magnetized by external source. Cited references [61,62] utilize magnetic microstructures with length of micrometers and diameter of 35nanometers – i.e. significantly larger than “several nanometers”.)

Line 432: Recently, the micro excitation coils has also been studied. – Please provide direct citation of source (even in case of pre-print or just submitted paper).

Also, please consider reformatting the list of references to include source journal and/or DOI. It would greatly simplify looking up the original source material. I have inspected MDPI Biosensors to check their required formatting of citations and published articles in there normally provide references including journal name, as for example in one recent publication:

References

1.         El-Ali, J.; Sorger, P.K.; Jensen, K.F. Cells on chips. Nature 2006442, 403–411.

2.         Jiang, C.; Dong, L.; Zhao, J.; Hu, X.; Shen, C.; Qiao, Y.; Zhang, X.; Wang, Y.; Ismagilov, R.F.; Liu, S.; et al. High-Throughput Single-Cell Cultivation on Microfluidic Streak Plates. Appl. Environ. Microbiol. 201682, 2210–2218.

3.         Tehranirokh, M.; Kouzani, A.Z.; Francis, P.S.; Kanwar, J.R. Microfluidic devices for cell cultivation and proliferation. Biomicrofluidics 20137, 051502.

(Just an example).

There are some minor errors in English that can be easily corrected. As for the stylistic constructs, please consult a native English speaker or consider suggested changes.

Notes on typos and peculiar stylistic constructs:

Line 42: easy for three-dimensional detection - suggested: easy implementation of three-dimensional detection

Line 115: Hall sensors is able - Hall sensors are able

Line 117: … it is powerless to detect – … it is not able to detect

Line 177: conductive materials (such as Cu, Gr). – probably should be Cu, Cr for Chrome(?)

Line 244: However, these study are – should be: these studies are

Line 262: And it can be applied in special occasions – Try consulting a native English speaker, but in my view the word “occasions” should be replaced with “application areas”.

(Similarly, also in lines 308, 555, 606, 676).

Line 277: Because the signal is AC, this sensor outputs digital signals directly – do you mean: because the sensing part forms a resonant circuit with variable frequency of resonance? Then maybe it should be: Because the signal is AC, this sensor outputs digital (frequency) signals directly

Line 288: susceptible magnetic characteristics to environmental – probably should be: magnetic characteristics susceptible to environmental conditions

Line 295: volume of high-resolution fluxgate is large, which is difficult to miniaturize and electronic integration[93]. - volume of high-resolution fluxgate is large, it is difficult to miniaturize and does not facilitate easy electronic integration [93].

Line 332: Apply pressure … , and observe the output change… - suggested: The pressure was applied … , and the output was observed to change in X, Y and Z axes…

Line 338: The output of Figure 5, … - The output shown in Figure 5, …

Line 377: the strong magnetic strength  -  the strong magnetic field

Line 382: and easy miniaturization make magnetic particles have great potential – and easy miniaturization means that magnetic particles have great potential

Line 407: biased magnetic field - bias magnetic field

Line 463: Tito tomo - Tito Tomo

Line 466: Finger belly – for clarity: Finger muscle belly

Line 498 (in Fig. 12 caption): surfacethe in a - surface in a

Line 618: The blue line and yellow line – probably should be blue line and red line

Line 645: the waterproof of the device - the waterproofing of the device (Also in line 651)

Line 646: For piezoresistive or piezoelectric sensors, because there is current in the sensitive structure exposed to water, it is easy to be damaged.

– Suggested: …, because current flows in the sensitive structure exposed to water, it can be easily damaged.

Line 649: At this time, the magnetic sensor only needs … – In contrast, the magnetic sensor only needs …

Line 662 (Fig. 18 caption): the sensor is uesd to read - the sensor is used to read

Line 676: which makes these sensors have application potential

 – which makes these sensors applicable

or

- which makes these sensors suitable for application

Line 698: … put forward high requirements for the flexibility – consult a native English speaker, but I would rather use: … emphasize high requirements for the flexibility

Reviewer 2 Report

The paper is interesting and can be developed and other applications.

Reviewer 3 Report

In this review article, Man et al. systematically summarized research on magnetic tactile sensors. The topic is timely and the structure of the paper is well organized. It can be accepted after the following minor revision.

1.         In 2.1.1-2.1.5, even though the authors described the working mechanism of each type of magnetic sensor, illustrations should be provided to make the readers understand these working mechanisms easily.

2.         The numbers of reference papers should be added in the caption of figures to make readers easily locate the reference. For example, in Figure 2a, ……AMR sensor (design by Ping Yu [54]);

3.         For the figures reproduced from reference papers, the permission of copyright should be obtained from the original publisher.

Reviewer 4 Report

I think this is a very nice review manuscript on magnetic tactile sensors. The authors summarized the types of magnetic sensors, the source of the magnetic field and the structure very clearly, and the figures are attractive. However, the manuscript has some issues, which need to be fully addressed, before it can be considered for publication.

1.    The title and abstract mentioned ‘biomimetic’, but I can’t see any ‘biomimetic’ in the sensors and main text description.

2.    The authors need to add hydrogel magnetic sensors in the manuscript since hydrogel is biocompatible for e-skin and medical sensors.

3.    Page 1, line 43: ‘However, there is no article to systematically summarize researches on magnetic tactile sensors so far’. I don’t think so coz a lot of papers summarized this.
